# The Impact of HER2-Low Expression on Oncologic Outcomes in Hormone Receptor-Positive Breast Cancer

**DOI:** 10.3390/cancers15225361

**Published:** 2023-11-10

**Authors:** Woong Ki Park, Seok Jin Nam, Seok Won Kim, Jeong Eon Lee, Jonghan Yu, Jai Min Ryu, Byung Joo Chae

**Affiliations:** Division of Breast Surgery, Department of Surgery, Samsung Medical Center, Sungkyunkwan University School of Medicine, Seoul 06351, Republic of Korea; woongki.park@samsung.com (W.K.P.); seokjin.nam@samsung.com (S.J.N.); seokwon1.kim@samsung.com (S.W.K.); jeongeon.lee@samsung.com (J.E.L.); jonghan.yu@samsung.com (J.Y.); jaimin.ryu@samsung.com (J.M.R.)

**Keywords:** breast cancer, HER2-low, hormone receptor status, menopausal status, oncologic outcomes

## Abstract

**Simple Summary:**

Breast cancer, especially in Asia, is on the rise. With improvements in tailored treatment, needs for more individualized subtyping are increasing. This retrospective study aimed to compare human epidermal growth factor receptor 2 (HER2)-low and HER2-0 breast cancers in hormone receptor (HR)-positive patients and investigate how it affect patients. We reviewed over 10,000 breast cancer cases, focusing on strong HR-positive subtypes. HER2-low breast cancer patients tended to have better oncologic outcomes compared to HER2-0 patients. Importantly, this difference was more pronounced in postmenopausal patients. This research highlights the need for reevaluating how we classify breast cancer subtypes. Understanding these distinctions could potentially lead to more tailored treatments for patients with HER2-low breast cancer in the future.

**Abstract:**

Breast cancer is a prevalent malignancy with increasing incidence, particularly in Asian countries. Classification based on estrogen receptor (ER), progesterone receptor (PR), and human epidermal growth factor receptor 2 (HER2) status is pivotal in determining treatment. Recent advances have challenged the traditional dichotomy in HER2 classification, prompting investigation into the HER2-low subtype’s characteristics and outcomes. This retrospective study analyzed 10,186 non-metastatic hormone receptor (HR)-positive, HER2-negative breast cancer cases treated from 2008 to 2020. Data encompassed clinical, pathological, and treatment information. Oncologic outcomes included disease-free survival (DFS), overall survival (OS), and breast cancer-specific survival (BCSS). In total, 56.5% were HER2-low cases. Differences in patient characteristics were noted, with more *BRCA1/2* mutations and higher mastectomy rates in the HER2-low group (*p* = 0.002, *p* < 0.001, respectively). Fewer received adjuvant chemotherapy or radiation therapy, and fewer histologic and nuclear grade 1 tumors were identified (all *p* < 0.001). With a median follow-up of 64 months (range: 13–174), HER2-low cases exhibited better DFS, OS, and BCSS than HER2-0 cases (*p* = 0.012, *p* = 0.013, and *p* = 0.013, respectively). Notably, the prognosis differed between premenopausal and postmenopausal subgroups, with BCSS benefitting premenopausal patients (*p* = 0.047) and DFS and OS benefitting postmenopausal patients in the HER2-low group (*p* = 0.004, *p* = 0.009, respectively). Multivariate analysis confirmed HER2 status as an independent predictor of these outcomes (*p* = 0.010, *p* = 0.008, and *p* = 0.014, respectively). This extensive single-center study elucidates the favorable prognosis associated with HER2-low status in HR-positive breast cancer. However, this effect differs among premenopausal and postmenopausal patients, necessitating further research into the underlying tumor biology.

## 1. Introduction

Breast cancer is the most common cancer in women, with its incidence rapidly increasing, especially in Asian countries. Breast cancer is classified into four subtypes according to estrogen receptor (ER), progesterone receptor (PR), and human epithelial growth factor receptor 2 (HER2) expressions. HER2 status is classified according to the American Society of Clinical Oncology (ASCO)/College of American Pathologist (CAP) guidelines. Immunohistochemistry (IHC) scores of 0, 1+, and 2+ with in situ hybridization (ISH) negative are defined as HER2 negative, while IHC scores of 3+ and 2+ with ISH positive are defined as HER2 positive [1].

There have been outstanding improvements in breast cancer treatment since the introduction of anti-HER2-targeted drugs such as monoclonal antibodies, tyrosine kinase inhibitors (TKIs), and novel antibody–drug conjugates (ADCs) [2,3,4]. Until now, HER2 status has been categorized dichotomously when administrating targeted anti-HER2 treatments, which accounts for only approximately 15–20% of all breast cancers [5,6]. Whether targeted anti-HER2 treatment might also be effective in patients with HER2 IHC 1+, 2+ with ISH negative (referred to as HER2-low) has become a subject of significant interest in recent years. Notably, the phase 3 clinical trial DESTINY-Breast 04 demonstrated the efficacy of trastuzumab-deruxtecan (T-Dxd), an ADC, in the treatment of metastatic HER2-low breast cancer [7].

While several studies examining the prognostic impact of HER2-low expression have been published, the question of whether HER-low should be defined as a new subtype of breast cancer remains unresolved. Some studies, including a nationwide retrospective cohort, have reported significantly better survival in HER2-low patients compared to HER2-0 patients [8,9,10,11,12,13]. On the contrary, other studies have found no meaningful differences in oncologic outcomes between the two groups [14,15,16,17]. Thus, our study aimed to investigate the characteristics of HER2-low breast cancer compared to HER2-0 breast cancer, particularly in hormone receptor (HR)-positive breast cancer, and analyze the long-term oncologic outcomes.

## 2. Materials and Methods

### 2.1. Study Population

A retrospective analysis was conducted on non-metastatic HR positive breast cancer patients treated at Samsung Medical Center (SMC) from January 2008 to December 2020. Patient data were extracted from our institution’s prospectively maintained breast cancer clinical database. Inclusion criteria encompassed patients diagnosed with primary invasive breast cancer who tested positive for HR (defined as an Allred score of 6 or higher for ER and PR status in this study) and negative for HER2. Patients diagnosed with in situ carcinomas, phyllodes tumors, malignancies other than primary invasive breast cancers, and metastatic cancers were excluded. Additionally, patients with unknown HER2 status, inadequate data, or those lost to follow-up within 12 months of surgery were also excluded. In total, 10,186 patients were included in the analysis.

### 2.2. Data Collection

Data collection encompassed baseline patient clinical and pathological characteristics such as sex, age at diagnosis, initial menopausal status, *BRCA1/2* status, pathological TNM stage, nuclear grade (NG), histologic grade (HG), lymphovascular invasion (LVI) status, ER, PR status (including the Allred score), HER2 status, and Ki-67 index, and treatment administered (including the type of breast/axillary surgery, chemotherapy, radiation therapy, and endocrine therapy).

Anti-ER, anti-PR, and anti-HER2 monoclonal antibody performed on 10% formalin-fixed and paraffin-embedded tissue were used to interpret ER, PR, and HER status. Additionally, the total Allred score was calculated by summing the intensity score (ranging from 0 to 3) and proportion score (ranging from 0 to 5). The Ki-67 index was assessed as follows. After scanning the ki-67-stained slide at 200× magnification using a VENTANA iScan scanner (Ventana Medical Systems Inc., Tucson, AZ, USA), areas 2 to 5 were assessed with Ventana Virtuoso software (version 5.6; Ventana Medical Systems Inc., Tucson, AZ, USA). The count of positive cells was measured at 100–200× magnification. In cases of excised tissue, the area with the highest number of positive cells (referred to as the “hot spot”) within the tumor’s margin was measured, and the average value was calculated. For heterogeneous tumors or multiple tumors, separate ki-67 index values were obtained and reported. To determine the positive rate, at least 1000 cells were counted. If the measurable tumor cell count fell below 1000, all tumor cells on the slide were included in the count.

Chemotherapy regimens were administered in accordance with the National Comprehensive Cancer Network guidelines and Korean national health insurance policies. Up to the time of this study, ADC’s including T-Dxd were only approved for use in HER2-positive breast cancer patients. Consequently, none of the patients in this study received ADC treatment.

The menopausal status was recorded based on patient’s menstruation history and serum follicle stimulating hormone (FSH) level at the initial patient visit to our medical center. Postmenopausal status was defined as the absence of menstruation for at least 12 consecutive months and a serum FSH level of 30 mIU/mL or higher. Recurrence or metastasis status, as well as survival data, were also collected. All tumors were pathologically staged according to the 6th or 7th edition of TNM classification by the American Joint Committee on Cancer (AJCC). HER2 status was assessed by pathologists in SMC pathology department in accordance with the 2007, 2013, 2018 ASCO/CAP guidelines. Patients with an IHC score of 0 were classified as HER2-0, while patients with an IHC score of 1+, 2+(ISH-negative) were classified as HER2-low. Disease-free survival (DFS) was defined as the interval from the date of diagnosis to local-regional recurrence or distant metastasis. Overall survival (OS) was defined as the interval from the date of diagnosis to death from any cause or censored at the last follow-up date. Breast cancer-specific survival (BCSS) was defined as the interval from the date of diagnosis to death caused by breast cancer progression or censored at the last follow-up date.

### 2.3. Statistical Analysis

Categorical variables were analyzed using Chi-square test or Fisher’s exact test while continuous variables were analyzed using the student *t*-test. DFS, OS, BCSS were estimated using the Kaplan–Meier method and the *p*-values of the log-rank test were used to assess significance. Univariate and multivariate analyses were performed using the Cox regression model. Variables that were considered relevant or showed a *p*-value < 0.05 were included in the multivariate Cox regression model. All *p*-values were two-sided and a *p*-value < 0.05 was considered statistically significant in our study. All data analyses were performed using the SPSS statistical software program, version 28.0 (SPSS, Chicago, IL, USA).

### 2.4. Ethics

This study was approved by the Institutional Review Board (IRB) of our institution (IRB no. SMC 2023-09-125-001).

## 3. Results

### 3.1. Clinical-Pathological Characteristics

A total of 10,186 patients with HR-positive, HER2-negative breast cancer were analyzed. Among them, 4429 patients (43.5%) were classified as HER2-0 and 5757 patients (56.5%) were classified as HER2-low. The clinical-pathological characteristics of HER2-0 and HER2-low are summarized in Table 1.

There were no significant differences in age or menopausal status between the two groups. However, *BRCA1/2* mutation was more frequently found in HER2-low patients (*p* = 0.002). In terms of the type of breast and axillary surgery performed, the proportion of mastectomies and sentinel lymph node biopsies (SLNB) was higher in the HER2-low group compared to the HER2-0 group (*p* < 0.001). Fewer patients in the HER2-low group received adjuvant chemotherapy or radiation therapy (*p* < 0.001). There were no differences in the proportion of patients receiving neo-adjuvant chemotherapy (NAC) or the type of anti-hormonal therapy administered between the two groups. The pathological T and N stages did not differ significantly. However, the HER2-low group had fewer histologic-grade (HG) and nuclear-grade (NG) 1 tumors compared to the HER2-0 group (*p* < 0.001). The Ki-67 index was categorized with a cutoff of 20%, and there were no significant differences observed between the two groups.

### 3.2. Oncologic Outcomes according to HER2 Status

The median follow-up time for the entire study population was 64 months (range: 13–174). Significant differences were observed in DFS, OS, and BCSS between the HER2-0 and HER2-low group (*p* = 0.012, *p* = 0.013, *p* = 0.013, respectively) (Figure 1).

We conducted a subgroup analysis on both the premenopausal and postmenopausal patient group to investigate whether the differences in DFS, OS, and BCSS remained significant within each subgroup. In the premenopausal subgroup, no significant differences were observed in DFS and OS based on HER2 status. However, when examining BCSS, the HER2-low group had a notably improved outcome compared to the HER2-0 group (*p* = 0.047). Conversely, in the postmenopausal subgroup, the HER2-low group exhibited significantly better DFS and OS (*p* = 0.004, *p* = 0.009, respectively). There were no notable differences in BCSS (Figure 2).

Given the notable differences in survival outcomes based on menopausal status, we conducted an analysis of the baseline characteristics of postmenopausal patients according to HER2 status (Appendix A). However, there were no significant differences when compared to the baseline characteristics of the overall study population.

### 3.3. Factors Associated with Oncologic Outcomes

To identify the factors associated with oncologic outcomes, univariate and multivariate analyses were performed (Table 2). In the univariate model for DFS, several factors, including well-established prognostic factors, were significantly linked to DFS outcomes. Notably, HER2 status (HR = 0.793, *p* = 0.014), menopausal status (HR = 0.769, *p* = 0.011), the type of breast and axillary surgery (HR = 2.198, *p* < 0.001 and HR = 2.651, *p* < 0.001, respectively), adjuvant chemotherapy and radiation therapy (HR = 1.493, *p* < 0.001 and HR = 1.547, *p* < 0.001, respectively), and the presence of *BRCA1/2* mutations (HR = 2.404, *p* < 0.001) were associated with DFS outcomes. In the multivariate model, HER2 status (HR = 0.745, *p* = 0.010) and the presence of *BRCA1/2* mutations (HR = 3.023, *p* < 0.001) remained significant predictors of DFS. Additionally, chemotherapy (HR = 0.651, *p* = 0.004), radiation therapy (HR = 2.776, *p* < 0.001), pT, pN stages, as well as HG remained significant predictors of DFS. Other factors did not maintain significant associations.

In the univariate model for OS, HER2 status (HR = 0.730, *p* = 0.02), menopausal status (HR = 1.998, *p* < 0.001), axillary surgery (HR = 4.214, *p* < 0.001), and pT, pN stages were associated with OS outcomes. In the multivariate model, HER2 status (HR = 0.630, *p* = 0.008), menopausal status (HR = 2.362, *p* < 0.001), the presence of *BRCA1/2* mutations (HR = 3.300, *p* = 0.01), axillary surgery (HR = 2.091, *p* = 0.003), adjuvant chemotherapy and radiation therapy (HR = 0.373, *p* < 0.001 and HR = 2.534, *p* < 0.001), higher pT and pN stages, and Ki-67 expression exceeding 20% (borderline significance, *p* = 0.057) remained as significant predictors of OS.

Regarding BCSS, HER2 status (HR = 0.553, *p* = 0.014), type of breast and axillary surgery (HR = 3.569, *p* < 0.001 and HR = 9.065, *p* < 0.001), and chemotherapy (HR = 3.082, *p* < 0.001), as well as pT and pN stages, HG, NG, and Ki-67 index, were all associated with BCSS (all *p* < 0.001). In the multivariate model, HER2 status (HR = 0.462, *p* = 0.014) and axilla surgery (HR = 2.378, *p* = 0.060) remained significant predictors, while radiation therapy (HR = 2.552, *p* = 0.031) became significant as a predictor of BCSS. Other factors in the multivariate model did not maintain significant associations. Overall, HER2-low status was an independent factor associated with DFS, OS, and BCSS.

### 3.4. Subgroup Analysis of HER2-Low Postmenopausal Patients according to Adjuvant Endocrine Treatment

Given the known occurrence of treatment resistance to tamoxifen in HR-positive, HER2-positive breast cancers, we conducted a comparison of outcomes to determine whether this phenomenon persisted in HER2-low breast cancers. Adjuvant endocrine treatment was administered in two types within the postmenopausal subgroup. Postmenopausal patients were either treated with tamoxifen (TAMO) alone or aromatase inhibitor (AI) alone. In our analysis, no significant differences were observed in DFS, OS, or BCSS between TAMO- and AI-treated groups (*p* = 0.770, *p* = 0.241, and *p* = 0.423, respectively (Figure 3).

## 4. Discussion

Our study investigated the clinical-pathological differences and long-term oncologic outcomes between HER2-low and HER2-0, HR-positive breast cancer patients. In our study, the proportion of HER2-low breast cancer was 56.5%, whereas previous studies reported proportions ranging from 31% to 79% [8,10,11,12,13,14,16,17,18]. Our results revealed that the HER2-low group had a higher proportion of patients who did not receive adjuvant chemotherapy or radiation therapy. Furthermore, fewer histologic and nuclear grade 1 tumors were observed in the HER-low group compared to the HER2-0 group. HER2-low breast cancer had significantly better outcomes in terms of DFS, OS, and BCSS. In multivariate analyses, HER2 status emerged as an independent predictor of DFS, OS, and BCSS.

Interestingly, the oncologic outcomes differed between the premenopausal and postmenopausal patients. Among premenopausal patients, only BCSS was significantly different according to HER2 status. On the other hand, there were notable differences in DFS and OS in postmenopausal patients.

The characteristics of HER2-low breast cancer differed variously in previous studies. A single-center retrospective study including 5235 stage I-III HER2-negative breast cancers reported that HER2-low breast cancers had more premenopausal patients compared to HER2-0 breast cancers [19]. In contrast, results published by Jacot et al. and Gampenrieder et al. indicated a higher proportion of older age patients in the HER2-low breast cancer group [20,21]. However, in our study, there were no notable differences in this regard.

When examining the pathological characteristics, Horisawa et al. reviewed 4007 HER2-0 and HER2-low patients and reported that HER2-low breast cancer had a lower T stage and a lower HG compared to HER2-0 breast cancer [16]. Additionally, a retrospective study including 351 node-negative breast cancer patients without adjuvant chemotherapy reported that HER2-low breast cancer had lower HG and lower Ki-67 index [13]. In a nationwide retrospective study of 30,491 Korean breast cancer patients, HER2-low breast cancer was characterized by a smaller tumor size, lower Ki-67 index, absence of LVI, but a higher HG when compared to HER2-0 breast cancer. A meta-analysis including 23 retrospective studies focusing on HER2-low found that HER2-low breast cancer had lower grade and lower T stage compared to HER2-0 breast cancer [10,12]. In general, previous studies suggest that HER2-low breast cancer tended to exhibit milder characteristics compared to HER2-0. However, in our study, no significant differences were observed in terms of menopausal status, pT stage, pN stage, Ki-67 index, and LVI. On the contrary, HER2-low breast cancer had a higher proportion of HG and NG 2 tumors, and, although the percentage was small, a higher detection rate of BRCA1/2 mutations compared to HER2-0 breast cancer. It is important to note that previous studies included all HER2-negative breast cancer patients and divided them into HER2-low and HER2-0 subgroups. In contrast, our study specifically selected strong HR-positive, HER2-negative patients for analysis. Therefore, the differences in baseline characteristics may be due to the slight variation in the study population.

The prognostic value of HER2-low was found to be significant in our study. However, several studies, including a retrospective study of 2605 breast cancer patients, have concluded that HER2-low status has no prognostic value on oncologic outcomes [14,16,22]. On the other hand, many studies, with larger sample sizes, have shown that HER2-low is a significant prognostic factor [8,10,11,12,13,18]. Studies reporting a significant difference in prognosis included studies with longer median follow-up period compared to studies that did not find any differences in oncologic outcomes. Therefore, the main cause of difference in outcomes may largely be due to late recurrence or mortality.

The mechanism behind why HER2-low breast cancer exhibits better outcomes compared to HER2-0 breast cancer remains unclear. Tan et al. described that the better recurrence-free survival and OS of HER2-low were mainly driven by the HER2 IHC1+ subgroup rather than the HER2 IHC2+, ISH-negative subgroup [8]. Zhang et al. re-evaluated 281 breast cancer cases diagnosed at a single institution and observed inter-observer variations in evaluating HER2 IHC staining, particularly in IHC 0 and IHC 1+ [23]. Similarly, Lambelin et al. reported inter-observer variation in up to 75% of the cases that were reassessed [24]. The low reproducibility of HER2 IHC results is a major obstacle in clearly defining HER2-low.

Additionally, Agostinetto et al. revealed a correlation between *ERBB2* expression and HER2 IHC scores, and Tan et al. also reported a significant difference in *ERBB2* expression levels based on HER2 IHC scores and *ERBB2* copy number variant (CNV) scores [8,14]. Due to the retrospective nature of our study, there were insufficient data regarding *ERBB2* expression or *ERBB2* CNV scores. Future studies on HER2-low breast cancer should investigate characteristics according to biomarker expressions. In summary, a new modality that includes these biomarker expression tests may help assess HER2 status more comprehensively.

To the best of our knowledge, our study is the largest single-center study regarding the oncologic outcomes of HR-positive HER2-low breast cancer. Moreover, we extracted data from our prospectively collected database, ensuring the homogeneity of treatment applied. Additionally, we only included patients with strong HR-positive status (an Allred score of 6 or higher), thus minimizing the influence of other tumor subtypes on outcomes. This study design made it possible to compare and understand the pure impact of HER2-low on breast cancer prognosis. In comparison to the previous studies, we discovered that the prognostic impact of HER2-low expression differed according to menopausal status.

The exact cause of these findings remains clear, and further studies comparing the different characteristics according to HER2-low, including menopausal status, are warranted. In the Kaplan–Meier curves comparing the HER2-low and HER2-0 groups, the survival gap between the two groups appeared to increase as the follow-up period extended. Given the risk of late recurrence in HR-positive breast cancer, previous results reporting no survival difference may be due to a shorter follow-up period.

Our study does have limitations. Since it was a non-randomized retrospective study, it included incomplete patient data, which introduces the possibility of other confounding variables, whether measured or unmeasured. Additionally, this study included only non-metastatic invasive strong HR-positive/HER2-0 or HER2-low Korean patients. Therefore, our results cannot be generalized to the entire breast cancer population or on a global scale. The study included patients from 2008 to 2020; a period during which treatment guidelines and adjuvant treatments evolved, potentially affecting the results. Furthermore, the ASCO/CAP guidelines for interpretating HER2 status were updated twice during this period. Patients classified as a HER2 IHC score 1+ according to 2007 ASCO/CAP guidelines would have been classified as HER2 IHC score 0 since the implementation of the updated 2013 and 2018 ASCO/CAP guidelines. In this study, a total of 889 patients were classified as HER2 IHC score 1+ during the 2008–2013 period, constituting 8.7% of the total study population.

## 5. Conclusions

In conclusion, we discovered that HER2-low status is an independent factor associated with a favorable prognosis in the context of strong HR-positive breast cancer. However, the prognostic value varied according to menopausal status. For HR-positive/HER2-0 postmenopausal patients, routine systemic imaging studies in conjunction with locoregional check-ups to detect late recurrence and/or distant metastases could prove beneficial. Future studies comparing tumor biology between HER2-low and HER2-0 are required.

## Figures and Tables

**Figure 1 cancers-15-05361-f001:**
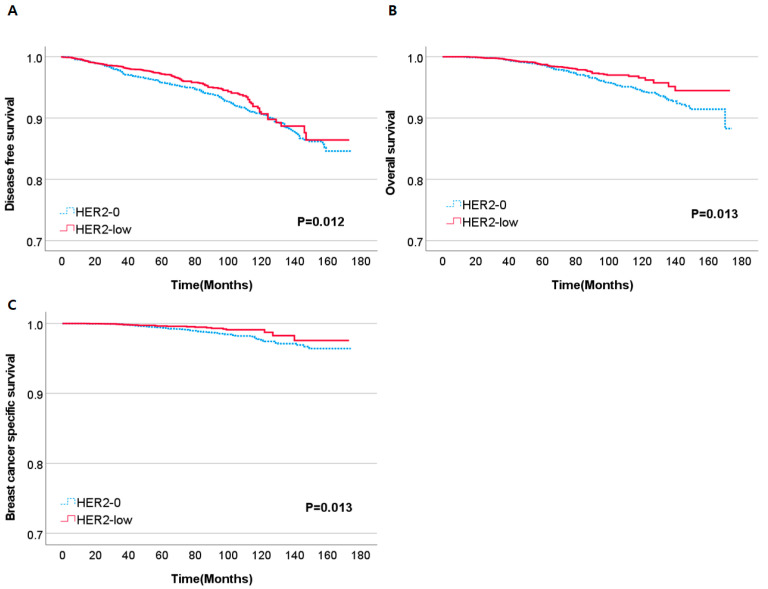
Kaplan–Meier curves of DFS (**A**), OS (**B**), BCSS (**C**) according to HER2 status. Abbreviations: DFS, disease-free survival; OS, overall survival; BCSS, breast cancer-specific survival.

**Figure 2 cancers-15-05361-f002:**
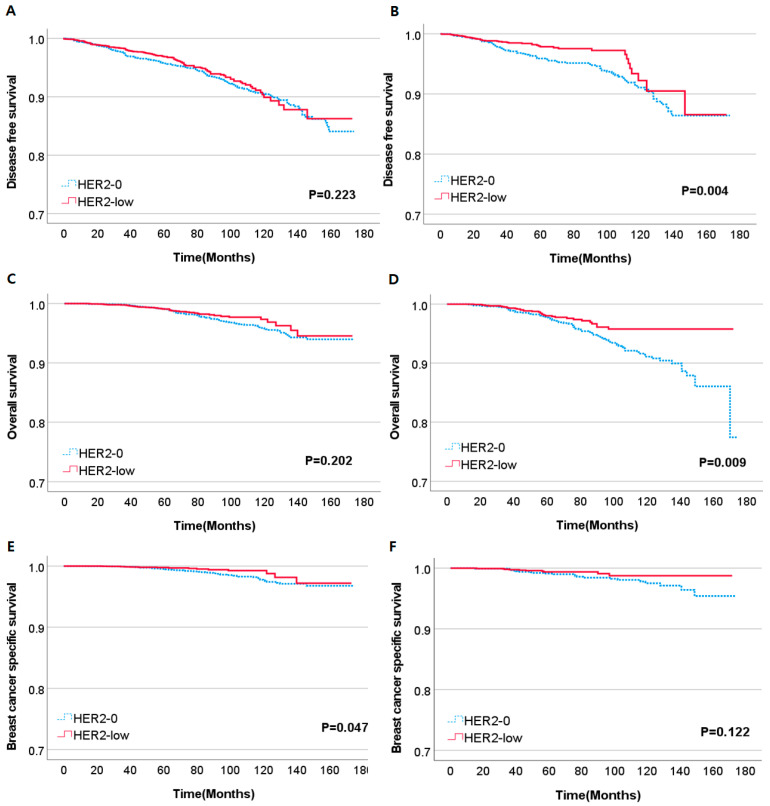
Kaplan–Meier curves of DFS, OS, BCSS according to HER2 status. (**A**,**C**,**E**) are DFS, OS, BCSS curves of premenopausal subgroup and (**B**,**D**,**F**) are DFS, OS, BCSS curves of postmenopausal subgroup.

**Figure 3 cancers-15-05361-f003:**
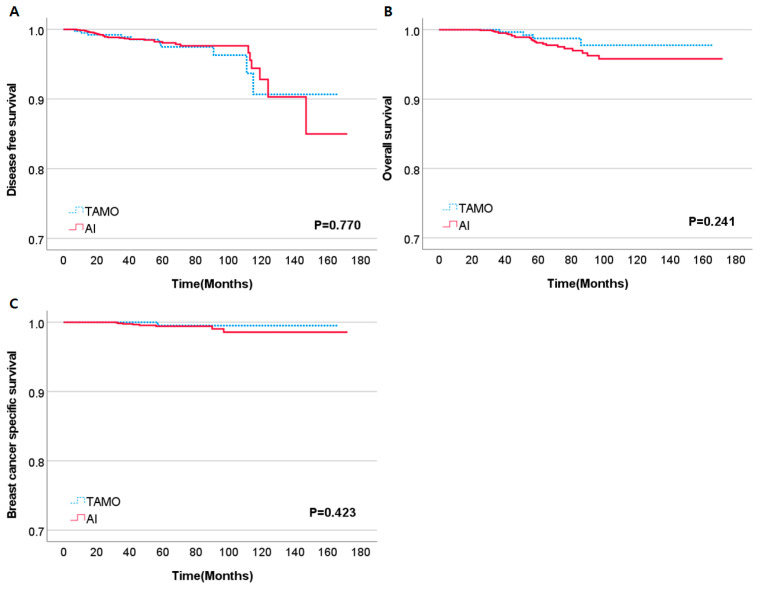
Kaplan–Meier curves of DFS (**A**), OS (**B**), BCSS (**C**) according to endocrine treatment in HER2-low, postmenopausal patients. Abbreviations: TAMO, Tamoxifen; AI, Aromatase inhibitor.

**Table 1 cancers-15-05361-t001:** Clinico-pathological characteristics according to HER2 expression.

	HER2 Status		*p* Value
	HER2-0	HER2-Low	
**Clinical Variables**			
Total no.	4429	5757	
Age			
Mean ± SD	48.84 ± 9.85	49.07 ± 9.68	0.236
Median(range)	47 (22–86)	48 (20–86)	
Sex, no. (%)			0.815
Male	23 (0.5%)	28 (0.5%)	
Female	4406 (99.5%)	5729 (99.5%)	
Menopausal status, no. (%)			0.125
Pre	2991 (67.9%)	3807 (66.4%)	
Post	1415 (32.1%)	1923 (33.6%)	
*BRCA1/2* mutation, no. (%)			0.002
Not detected	4384 (99.0%)	5657 (98.3%)	
Detected	45 (1.0%)	100 (1.7%)	
Breast surgery, no. (%)			<0.001
BCS	3243 (73.2%)	3750 (65.1%)	
Mastectomy	1186 (26.8%)	2007 (34.9%)	
Axillary surgery, no. (%)			<0.001
No axillary surgery	15 (0.3%)	17 (0.3%)	
SLNB	2895 (65.4%)	4108 (71.4%)	
ALND	1519 (34.3%)	1632 (28.3%)	
Adjuvant chemotherapy, no. (%)			<0.001
Not done	2455 (55.7%)	3628 (63.2%)	
Yes	1952 (44.3%)	2112 (36.8%)	
Neo-adjuvant chemotherapy, no. (%)			0.631
No	4224 (95.4%)	5502 (95.6%)	
Yes	205 (4.6%)	255 (4.4%)	
Radiation therapy, no. (%)			<0.001
No	811 (18.5%)	1048 (24.5%)	
Yes	3573 (81.5%)	4332 (75.5%)	
Anti-hormonal therapy, no. (%)			0.282
Tamoxifen	3223 (74.2%)	4068 (73.2%)	
Aromatase inhibitor	1122 (25.8%)	1488 (26.8%)	
**Pathological variables**			
* T stage			0.271
1	2831 (67.0%)	3594 (65.3%)	
2	1230 (29.1%)	1687 (30.7%)	
3	154 (3.6%)	213 (3.9%)	
4	9 (0.2%)	8 (0.1%)	
* N stage			0.096
0	2769 (65.8%)	3688 (67.3%)	
1	1078 (25.6%)	1384 (25.2%)	
2	231 (5.5%)	284 (5.2%)	
3	128 (3.0%)	126 (2.3%)	
Histologic grade			<0.001
1	1687 (38.4%)	1753 (30.7%)	
2	2302 (52.4%)	3413 (59.8%)	
3	405 (9.2%)	543 (9.5%)	
Nuclear grade			<0.001
1	1084 (24.6%)	835 (14.5%)	
2	2837 (64.3%)	4290 (74.7%)	
3	488 (11.1%)	617 (10.7%)	
Ki-67, %			0.960
<20%	2699 (71.0%)	3339 (71.1%)	
≥20%	1102 (29.0%)	1360 (28.9%)	
Lymphovascular invasion			0.384
No	3120 (70.4%)	4101 (71.2%)	
Yes	1309 (29.6%)	1656 (28.8%)	

* 460 patients who received neo-adjuvant chemotherapy were not included.

**Table 2 cancers-15-05361-t002:** Univariate, multivariate Cox models regarding DFS, OS, BCSS.

Disease-Free Survival				
	Univariate Model HR (95% CI)	*p* Value	Multivariate Model HR (95% CI)	*p* Value
HER2 status (0 vs. low)	0.793 (0.659–0.953)	0.014	0.745 (0.594–0.933)	0.010
Menopause (Pre vs. Post)	0.769 (0.628–0.941)	0.011	0.884 (0.697–1.120)	0.307
*BRCA1/2* mutation (no vs. yes)	2.404 (1.461–3.956)	<0.001	3.023 (1.721–5.308)	<0.001
Breast Surgery (BCS vs. TM)	2.198 (1.843–2.622)	<0.001	0.703 (0.493–1.005)	0.053
Axilla Surgery	2.651 (2.214–3.174)	<0.001	1.067 (0.772–1.474)	0.694
Chemotherapy (no vs. yes)	1.493 (1.244–1.791)	<0.001	0.651 (0.486–0.873)	0.004
Radiation therapy (yes vs. no)	1.547 (1.270–1.885)	<0.001	2.776 (1.901–4.055)	<0.001
pT stage	2.130 (1.871–2.425)	<0.001	1.720 (1.424–2.078)	<0.001
pN stage	1.785 (1.622–1.963)	<0.001	1.563 (1.313–1.861)	<0.001
Histologic grade	2.071 (1.808–2.371)	<0.001	2.008 (1.526–2.642)	<0.001
Nuclear grade	1.797 (1.546–2.090)	<0.001	0.859 (0.641–1.153)	0.311
Ki-67 (<20% vs. >20%)	2.119 (1.738–2.584)	<0.001	1.247 (0.982–1.583)	0.070
**Overall Survival**				
HER2 status (0 vs. low)	0.730 (0.560–0.952)	0.02	0.630 (0.447–0.887)	0.008
Menopause (Pre vs. Post)	1.998 (1.554–2.569)	<0.001	2.362 (1.728–3.230)	<0.001
*BRCA1/2* mutation (no vs. yes)	1.678 (0.746–3.772)	0.21	3.300 (1.333–8.172)	0.01
Breast Surgery (BCS vs. TM)	2.724 (2.126–3.490)	<0.001	0.800 (0.502–1.273)	0.346
Axilla Surgery	4.214 (3.192–5.564)	<0.001	2.091 (1.290–3.389)	0.003
Chemotherapy (no vs. yes)	1.235 (0.954–1.597)	0.109	0.373 (0.247–0.562)	<0.001
Radiation therapy (yes vs. no)	1.675 (1.270–2.208)	<0.001	2.534 (1.535–4.181)	<0.001
pT stage	2.636 (2.220–3.131)	<0.001	2.100 (1.619–2.723)	<0.001
pN stage	2.053 (1.812–2.326)	<0.001	1.541 (1.216–1.952)	<0.001
Histologic grade	1.995 (1.655–2.406)	<0.001	1.269 (0.869–1.855)	0.217
Nuclear grade	2.052 (1.669–2.523)	<0.001	1.171 (0.786–1.746)	0.437
Ki-67 (<20% vs. >20%)	2.115 (1.595–2.805)	<0.001	1.407 (0.990–1.999)	0.057
**BC-Specific Survival**				
HER2 status (0 vs. low)	0.553 (0.344–0.889)	0.014	0.462 (0.249–0.857)	0.014
Menopause (Pre vs. Post)	1.307 (0.848–2.016)	0.225	1.527 (0.888–2.628)	0.126
*BRCA1/2* mutation (no vs. yes)	2.396 (0.757–7.577)	0.137	2.870 (0.678–12.143)	0.152
Breast Surgery (BCS vs. TM)	3.569 (2.336–5.453)	<0.001	0.848 (0.402–1.787)	0.664
Axilla Surgery	9.065 (5.026–16.350)	<0.001	2.378 (0.964–5.864)	0.06
Chemotherapy (no vs. yes)	3.082 (1.860–5.104)	<0.001	1.495 (0.534–4.183)	0.444
Radiation therapy (yes vs. no)	1.181 (0.717–1.947)	0.514	2.552 (1.091–5.970)	0.031
pT stage	4.355 (3.309–5.732)	<0.001	3.060 (2.009–4.661)	<0.001
pN stage	2.576 (2.094–3.169)	<0.001	1.326 (0.912–1.928)	0.139
Histologic grade	3.238 (2.349–4.463)	<0.001	2.107 (1.103–4.027)	0.024
Nuclear grade	3.395 (2.384–4.835)	<0.001	1.188 (0.597–2.364)	0.623
Ki-67 (<20% vs. >20%)	3.457 (2.148–5.562)	<0.001	1.444 (0.817–2.552)	0.206

Abbreviations: DFS, disease-free survival; OS, overall survival; BCSS, breast cancer-specific survival; BCS, breast conserving surgery; TM, total mastectomy.

## Data Availability

Research data that support the findings of this study are securely stored in an institutional repository and are available to share from the corresponding author upon reasonable request.

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
