# Peer review of "The Impact of HER2-Low Expression on Oncologic Outcomes in Hormone Receptor-Positive Breast Cancer"

_cancers, 2023, doi:10.3390/cancers15225361_

Round 1

Reviewer 1 Report

Comments and Suggestions for Authors

This is an important issue and the authors should be commended on their work.  One area that they might consider discussing further is the literature regarding cross-talk between ER and HER2 

The authors are addressing the question of differences in outcome between HER2 0 and HER2 low breast cancer patients as measured by disease free survival, disease specific survival and overall survival. They have examined a large population in South Korea in a retrospective analysis.

While the topic is not original, it is quite important. Recent studies have shown that the antibody drug conjugate Trastuzumab deruxtecan, which was developed for HER2 positive disease, is also active in HER2 low disease.  However, there is insufficient data at this point to know whether or not there is an outcome difference between HER2 0,1+ and 2+ (FISH negative) patients with this drug. Regardless of the answer to that question, it does raise the issue of whether or not there is biologic difference between HER2 0 and “HER2 low” patients.  Multiple papers have attempted to examine this. In that regard, this manusciprt is not uinique.  However, it does have a very large patient cohort, which is helpful. The discussion also references the other studies that have been done in this area, highlighting the lack of unanimity of those studies in attempting to answer this question.

It adds a very large cohort compared to most of the other studies in the area. The data on differences in outcome by menopausal status is a new observation.

This study lacks detailed information on hormone receptor status. There is a significant body of literature addressing “crosstalk” between estrogen receptor and HER2. It would be helpful for the authors to review, to some degree, this material. It would have been useful to have quantitative data on the degree of hormone receptor positivity (including both ER and PR) and how that impacts the outcomes they are assessing. In addition, they only looked at HR positive patients. They did not define HR positive, which would be useful. In addition, the field still lacks adequate analysis of HR negative, HER2 low and HER2 0 patients. It would have been of interest for the authors to have also looked at this population. Finally, it would be of interest, given the size of the cohort, to look at what kinds of chemotherapy were given – particularly whether or not patients receivd T-Dxd and/or other ADC’s.

Author Response

Thank you very much for taking the time to review this manuscript. Please see the attachment for detailed replies to your comments.

Reviewer 2 Report

Comments and Suggestions for Authors

The study investigated the clinical-pathologic features and long-term outcomes of HER2-low hormone receptor positive breast cancer compared to HER2-negative breast cancer. The following revisions are recommended.

- Line 54: Define ISH.

- Lines 78-80: Change "Patients diagnosed with primary invasive breast cancer with a strong hormone receptor positive (defined as an Allred score of 6 or higher in ER, PR status in this study) and HER2-negative state were included" to "Patients diagnosed with primary invasive breast cancer, positive for hormone receptors (defined as an Allred score of 6 or higher in ER, 79 PR status in this study) and negative for HER2, were included.

- Line 80: Add the following information: a) ER, PR and HER2 antibodies and immunohistochemical protocol; b) Specify how Allred score was calculated.

- Line 89: Add information about the Ki-67 clone and how it was scored.

- Line 88: Change "lympho-vascular" to "lymphovascular".

- Line 92: Define FSH.

- Use "pathologic" (line 117) or "pathological" (e.g. line 86) consistently throughout the manuscript.

- Line 122, 164, Tables 1-2 and anywhere else in the manuscript: Italicize gene names - BRCA1/2.

- Table 1: a) What is the median age and the range? b) Change "Axilla surgery" to "Axillary surgery".

Comments on the Quality of English Language

Satisfactory

Author Response

(The authors gave the same response as above.)

Reviewer 3 Report

Comments and Suggestions for Authors

This retrospective study delves into the impact of HER2-Low expression on oncological outcomes in hormone receptor-positive breast cancer. It meticulously analyzes data from 10,186 patients with hormone receptor-positive and HER2 negative breast cancer, juxtaposing HER2-Low with HER2-0 subtypes. The research methodologies encompass both univariate and multivariate analyses, and a robust statistical analysis leveraging the Chi-square test, Fisher’s exact test, student t-test, Kaplan-Meier method, and Cox regression model. A salient conclusion drawn is that patients with HER2-Low breast cancer manifest better prognostic outcomes compared to their HER2-0 counterparts. This study illuminates the nuanced differences between HER2-Low and HER2-0 breast cancers in HR+ patients and offers significant insights into the ramifications of HER2-Low expression on oncological outcomes.

However, I have some pivotal critiques which I believe should be addressed to enhance the quality of the manuscript:

Given that this research hinges on the differentiation between HER2-low and HER2-negative, it is of paramount importance to lucidly delineate the criteria defining these two categories. This would facilitate future validation or comparison with data from other sources.

The manuscript notes that the "HER2 Status was assessed by the pathologist according to the 2007, 2013, 2018 ASCO/CAP guidelines." It's imperative to ascertain if these guidelines remained consistent over the years. If discrepancies exist, how was the classification harmonized to a singular standard?

To fortify the study, a more comprehensive discourse on the potential clinical ramifications of the findings would be beneficial. The authors might consider elucidating further on the practical applications of their discoveries and their potential influence on patient outcomes.

In summation, I am of the conviction that this study bears significant scientific relevance and furnishes invaluable insights underscoring the significance of individualized subtyping in breast cancer treatment.

Comments on the Quality of English Language

minor edits needed

Author Response

(The authors gave the same response as above.)

Round 2

Reviewer 2 Report

Comments and Suggestions for Authors

All abbreviations should be defined at first use - that includes ISH on line 54, NCCN on line 104.

Comments on the Quality of English Language

NA

Author Response

Dear Reviewer 2,

We deeply thank you for the time and effort in reviewing this article. 

We appreciate your kind comment and totally agree with you.

Comment 1. All abbreviations should be defined at first use - that includes ISH on line 54, NCCN on line 104.

->Response to comment. 

We define all abbreviations including in situ hybridization (ISH) (line 54), National Comprehensive Cancer Network (NCCN) (lines 108-109), and hormone receptor (HR) line 74.